# An Eye Movement Study in Unconventional Usage of Different Software Tools

**DOI:** 10.3390/s23083823

**Published:** 2023-04-08

**Authors:** Jozsef Katona

**Affiliations:** 1CogInfoCom Based LearnAbility Research Team, Department of Software Development and Application, Institute of Informatics, University of Dunaujvaros, 2400 Dunaujvaros, Hungary; katonaj@uniduna.hu or katona.jozsef@kvk.uni-obuda.hu or Jozsef.Katona@hvl.no; Tel.: +36-25-551-605; 2Department of Instrumentation and Automation, Institute of Electronics and Communication Systems, Kandó Kálmán Faculty of Electrical Engineering, Obuda University, 1034 Budapest, Hungary; 3Department of Computer Science, Electrical Engineering and Mathematical Sciences, Faculty of Engineering and Science, Western Norway University of Applied Sciences, 5063 Bergen, Norway

**Keywords:** XAML, classic C#, cognition load, eye-tracking, programming

## Abstract

One of the main challenges of Human-Computer Interaction is the creation of UIs that enable the use of different systems in an easy and understandable method. The study analyses the student audience who uses software tools differently from the basis. In the research, two languages supporting UI implementation related to .NET technology, XAML and classic C#, were compared in terms of the cognitive load of test subjects. The results of the traditional knowledge level assessment tests and the answers to the questionnaires show that the UI implementation described in XAML is easier to read and understand than the same description in classic C#. When viewing the source codes, the eye movement parameters of the test subjects were also recorded and then evaluated, where a significant difference in the number and duration of fixations was observed, i.e., the interpretation of the classic C# source code showed a larger cognitive load. Overall, the results of the eye movement parameters supported the results of the other two measurement methods when comparing the different types of UI descriptions. The results established in the study and its conclusion may have an impact on programming education as well as industrial software development in the future, and also clearly shows the importance of choosing the development technology that best suits the person or development team.

## 1. Introduction

Software development has now become a critically complicated and complex process, as the production of source codes to be implemented, and later improved and maintained, is an increasingly difficult task. In order to facilitate the process, developers come up with new techniques, technologies, methods, tools, and paradigms.

Although GUI development environments are available today that effectively support development in this direction without even writing a single line of source code, there are basically two problems with this kind of approach. The first is that those students who, for example, only put together the graphic interfaces using the drag&drop method often do not understand what kind of code is generated in the background. The second is that, in most cases, these methods pollute the source code and generate more code than is absolutely necessary. This makes the code harder to understand and read and ultimately maintain. The GUI interface can be implemented at the source code level in several languages and tools.

One such tool is XAML (eXtensible Application Markup Language), which is an XML (Extensible Markup Language)-based declarative markup language that greatly simplifies the creation of user interfaces (UI). The language plays a particularly important role in the .NET framework, and it can be especially important in the development of WPF (Windows Presentation Foundation) or Xamarin applications, where the UI data elements, data bindings, and other services, etc. defines. In addition to all this, every XAML element type can be matched to a .NET class and the declarative description to the imperative code, the consequence of which is that everything we describe in XAML can also be described with classic C# code, so we can also dynamically create UI controls. However, creating an entire window completely in classic C# can result in much more complex and difficult-to-understand code. In WPF or Xamarin, XAML allows UI design and business logic to be managed or handled separately, resulting in much more readable code [1].

Figure 1 shows a semantically identical but different source code in the used language.

However, based on industrial and educational feedback, choosing the right toolkit is not always a clear task due to, among other things, the different cognitive abilities characteristic of individuals [2,3]. An objective examination of programming as a complex cognitive activity [4] can contribute to the implementation and implementation of more effective applications and software systems, as well as to the selection of the toolbox that best suits developers [5]. Eye movement tracking systems, which can be classified as human-computer interfaces, make it possible to objectively examine the readability and comprehensibility of the source code of software, as well as the overall cognitive load, by analyzing the path of the human gaze.

Recently, more and more research has been published that aims to reduce the costs of software development phases by using eye movement tracking systems since the observation and study of the gaze path are suitable for the analysis of more complex cognitive processes. A group of researchers studies the planning phase of the development cycle. Within the group, some of the visual patterns used to design UML class diagrams that provide a framework for object-oriented programs, as well as the cognitive effect of their application [6,7,8], while others analyze the layout of these diagrams and the resulting more effective comprehensibility and readability [9,10,11]. Another group of research analyses the product of the implementation activity, i.e., the readability, comprehensibility, and complexity of the created source codes [12,13]. In a study by Crosby & Stelovsky [14], they analyzed how subjects read a source code or text in their native language. As a result of the research, it was stated that a significant difference could be shown in the way of reading and reviewing the two types of text because, in the case of program codes, longer recording times were present. Crosby, Schlotz & Wiedenbeck [15] observed that novice programmers put much more emphasis on reading comments and additional information than their more experienced.

The aim of the research is to examine which language can be used to create a more understandable, readable, and at the same time, maintainable UI source code. In addition to the comprehension questions, the eye movement parameters of the test subjects were recorded and evaluated in order to reveal possible associations. In Section 2, the article provides a short literature review of the WPF technology, along with the application possibilities of the classic C# and XAML languages. In Section 3, the tools, materials, and methods used for the research are described in more detail. Section 4 and Section 5 evaluate the results of the research and formulate the discussions that can be drawn from the determined results. A shorter conclusion can be read in the last section.

## 2. Theoretical Background

In the research of Zhang & Liu [16], WPF technology was used in the implementation of the intelligent, client-server-based system for automatic battery bag detection, which is the basis of their research. The developed system is based on a three-layer (Presentation, Logic, and Data) architectural design model, so the UI was separated from the business logic of the application, and the results recorded and determined during the measurements were stored in a MySQL-based database and made exportable for carrying out further tests.

Kozminski [17] examined how important it is to design a user-friendly UI during a possible emergency intervention. A well-designed UI can reduce operator errors and test times while increasing efficiency. The article also discusses the WPF technology, which enables more intuitive, easier-to-use, and higher-quality UI development in a shorter time.

Wang et al. [18] developed a productivity-enhancing office automation (OA) system for small and medium-sized companies. After getting to know the specific work and management processes of companies, the functional and non-functional requirements of an application can be more easily explored. The authors used WPF technology for the software development, which is the basis of the research, as it supported the tools that increase the efficiency of the development, such as functional module design, multi-level architecture implementation, etc. The new OA system provides ease of use and improves the level of information management.

Lew et al. [19] showed that operators of critical processes have to deal with the continuous challenges of complex systems. During the research, it was stated that the development of human-machine interfaces (HMI) is of particular importance from the point of view of the safety and reliability of the operation. There are very few tools available for the efficient design and implementation of HMIs, so the authors formulated suggestions during their investigation that, based on WPF technology, can be well-matched and adapted to the HMI research and development required for process control.

Belenesi et al. [20] compare the possibilities of the UWP (Universal Windows Platform) framework published in recent years with WPF. Based on the results, it can be stated that an important factor of a framework can be the target device for which the application is developed.

During the research of Filipova-Petrakieva & Shopov [21], a desktop application related to data security was developed, the main purpose of which is to create educational software presenting the coding and decoding of information. Software tools such as C#, .NET Framework, WPF and XAML, and Entity Framework were used to implement the application. Based on the obtained results, it can be stated that the implemented software can effectively present the operation of the different encryption algorithms.

Guzsvinecz et al. [22] developed a desktop application using WPF for human motion classification. By designing an easy-to-use GUI, their aim was to support telerehabilitation on home computers with low-cost sensors such as the Microsoft Kinect. Their algorithm is able to predict repeating—but possibly changing—gestures of the patients without the use of machine learning methods.

In their research, Zhang & Ruan [23] implemented the online monitoring of gantry cranes based on the WPF platform and in a virtual reality (VR) environment. Since WPF effectively supports vector graphic development, it serves as a good basis for 3D dynamic monitoring of devices and equipment. By using stress analysis, the system can predict the safety risk in the steel structure in a predictive manner, so it can determine the condition of the gantry cranes.

The short literature background research supports the applicability and importance of WPF technology in the design and development of user-friendly UI. In addition to the classic C# language, the technology also makes XAML available to developers to implement this type of interface.

If we want to somehow measure the cognitive load of performing a task, we can measure test-related performance indicators [24], reveal subjective opinions using questionnaires [25], or use non-invasive psychophysical tools [26].

Nowadays, different tools are available that can measure cognitive load in real time [27,28,29]. With the help of eye movement tracking devices, we can record parameters such as fixation, saccade, or pupil diameter [30], and these parameters can also provide information about the attention level and cognitive status of a test subject [31].

Based on the above, the following research questions were formulated:

Research Question 1 [RQ1]: Can a significant difference be detected in the results of the tests with regard to the two languages?

Research Question 2 [RQ2]: Can a significant difference be detected in the visual parameters of the tests with regard to the two languages?

## 3. Materials and Methods

During the research, a total of 4 UI descriptions were used, which used elements of the same toolbox (e.g., buttons, labels, input fields, etc.), so the difficulty of their readability and their level of comprehensibility were the same. Of the 4 UI descriptions, 2 were written in XAML, while another 2 were written in classic C# in the Visual Studio development environment. In order to ensure that the difference in the knowledge levels of the test subjects does not affect the final result, a test subject had to interpret the UI written in both languages in each case. After studying the source codes, a knowledge-level assessment test examining the interpretability of the UI had to be completed. Finally, a subjective questionnaire consisting of a few questions examining readability and comprehensibility had to be filled out. The process describing the test is briefly summarized in Figure 2.

During the examination and interpretation of the source codes, the eye movement parameters of the test subjects were continuously recorded and subsequently evaluated using GazePoint 3 (GP3) (https://www.gazept.com/product/gazepoint-gp3-eye-tracker/, accessed on 1 March 2023) eye-tracker device and the OGAMA (http://www.ogama.net/, accessed on 1 March 2023) open-source software package. The device and the software package used are cost-effective, and several researchers have already used them successfully for previous research [32,33,34]. Figure 3a shows the GP3 research-grade eye tracker hardware unit, and Figure 3b the recording module of the OGAMA software package.

### 3.1. Test Subjects

The test was attended by 36 university students (10 woman and 26 man, age: *M* = 20.25 *SD* = 1.05), who successfully completed the subjects containing the study materials, the knowledge of which is essential for solving the tests. The test subjects volunteered for the test and declared themselves to be completely healthy. They were not under the influence of any medication and had no difficulties in reading or learning in the past and during this examination. Based on the results of the programming subject, test subjects with better-than-average programming skills and a similar level of knowledge were selected.

### 3.2. Test Conditions and Steps of the Research

The description of the UI interface as illustrated in the Visual Studio development environment on an LG22M45 22” monitor capable of 1920 × 1080 resolution. The GP3 unit was placed under the monitor at a distance of approximately 65 cm from the eyes of the test subjects. So that the test subjects are not affected by sudden changes in light, I naturally used uniform lighting. For each UI description, 1 XAML and 1 classic C# description were randomly selected for the test subjects. Since the density of images or texts can affect information processing [35], comprehensibility and easy readability are also important points in UI descriptions, so it was tried to set the same distance between the texts and the source codes. A total of 36 XAML and 36 classic C#-based eye movement parameter packages were saved in a database for further evaluation.

A schematic diagram of the testing environment is shown in Figure 4.

## 4. Results

In order to ensure a correct evaluation, it was considered that the same test subjects were examined within a group and that they were independent of each other. The normality of the variables was checked with the Shapiro-Wilk test. In the case of the tests, the *p* < 0.05 value was significant.

The evaluation of the results began with the evaluation of traditional knowledge-level assessment tests, which examined the comprehensibility and readability of the studied source code. Each test subject received the same 10 questions, so a total of 360 answers were evaluated.

### 4.1. The Evaluation of the Results of the Traditional Knowledge Level Tests

The test subjects were able to answer all questions regarding the comprehensibility and readability of the different types of source codes.

The results show that in the case of a UI written by XAML, in the worst case, the number of incorrect answers was a maximum of 3, which means 70% success, and in the best case, the test subjects answered all questions correctly. Regarding the total sample, *M* = 1.56, *SD* = 0.88 wrong answers were made, which shows an overall success rate of 84.4%.

In the case of the UI written with classic C#, the maximum number of errors was 4, which means a 60% success rate, and this method also had test subjects who were able to answer all questions correctly. Regarding the total sample, *M* = 2.53, *SD* = 1.40 wrong answers were made, which shows a success rate of 74.7% overall.

Table 1 summary table shows the number and percentage of incorrect answers in the knowledge level test regarding the XAML and classic C# code.

Figure 5 shows the distribution of the test subjects’ incorrect answers. Figure 5a is written in XAML, while Figure 5b is for a UI written in classic C#. In the case of XAML, there were four correct answers, 13 with one, 14 with two, and five with three errors, while in the case of classic C#, there were also four without errors, five with one, eight with two, six with three, and 13 with four errors.

The normality results of the traditional knowledge level tests measured are significant (XAML approach: *W*(36) = 0.879, *p* < 0.001, classic C# approach: *W*(36) = 0.857, *p* < 0.001). Due to the results of the normality test, the Wilcoxon signed-rank test was used: (XAML approach: *T* = −383, *Z* = 2.68, *p* = 0.007 (2-tailed), *r* = 0.316). Regarding the obtained results, it can be stated that the results of traditional knowledge level tests were significantly better with a small effect in the case of the XAML approach: *Mdn* = 2 than in the case of the classic C# approach: *Mdn* = 3.

### 4.2. The Evaluation of Test Subjects’ Subjective Opinion

After completing the traditional knowledge level assessment test, the test subjects also had to fill out a questionnaire consisting of six questions, which measured the subjective opinions regarding XAML and classic C#. During the survey, a 5-point Likert-type scale was used, where “A”: not at all; “B”: slightly; “C”: moderately; “D”: pretty; “E”: completely. The following questions were formulated in the questionnaire (Qs):

Q1: How difficult was it to understand the XAML code?

Q2: How difficult was it to understand classic C# code?

Q3: To what extent would you feel the need for comments for the XAML code?

Q4: To what extent would you feel the need for comments for classic C# code?

Q5: How much would you recommend XAML code for creating UI?

Q6: How much would you recommend classic C# code for creating UI?

After the evaluation, it can be seen that, according to the test subjects, the XAML code is easier to read and better understood without comments than the classic C# source code. In addition to all this, XAML code was much more recommended for creating UIs.

The result of the evaluation is shown in Figure 6.

Answering the RQ1 question, it can be stated that a significant difference can be shown (*p* = 0.007) between the results of the traditional knowledge level assessment tests. The results show that the UI description written in XAML is easier to understand and shows better test results than the classic C# source code that generates the same UI.

It can also be seen based on the subjective opinions that the test subjects preferred and easier to interpret the XAML-based codes, on the contrary, with the classic C# code.

### 4.3. Evaluation of Eye-Tracking Parameters

During the understanding and reading of the XAML and classic C# source codes, the many eye movement parameters of the test subjects were recorded, and after evaluation, a possible relationship between the individual parameters and the results of the knowledge level assessment tests may be shown.

#### 4.3.1. Fixation Duration Mean

The descriptive statistical table of the recorded fixation duration mean results can be read in Table 2 for XAML and classic C#.

Figure 7 shows the relationship between the fixation duration mean eye movement parameter recorded while reading and understanding the XAML and classic C# source codes.

The normality results of the fixation duration mean measured in the XAML code are significant (XAML: *W*(36) = 0.962, *p* = 0.251, C#: *W*(36) = 0.935, *p* = 0.036). Due to the results of the normality test, the Wilcoxon signed-rank test was used: (*T* = 496.5, *Z* = 2.57, *p* = 0.010 (2-tailed), *r* = 0.303). Regarding the obtained results, it can be stated that the fixation duration mean was significantly shorter with a medium effect in the case of the XAML: *Mdn* = 572 ms than in the case of the C#: *Mdn* = 670 ms.

The distribution of the fixation duration means based on XAML and classic C# are shown in Figure 8.

In response to the RQ2 question, a significant difference was found in the average of the fixation duration mean (*p* = 0.010) when the test subjects read and understood the source codes based on XAML or classic C#. The results show that a shorter fixation duration means, i.e., shorter information recording, was necessary in the case of the XAML-based UI description.

#### 4.3.2. Number of Fixations

The descriptive statistical table of the recorded number of fixations results can be seen in Table 3 for XAML and classic C#.

Figure 9 shows the relationship between the number of fixations eye movement parameter recorded while reading and understanding the XAML and classic C# source codes.

The normality results of the average fixation number measured in the XAML code are significant (XAML: *W*(36) = 0.935, *p* = 0.035, C#: *W*(36) = 0.951, *p* = 0.109). Due to the results of the normality test, the Wilcoxon signed-rank test was used: (*T* = 526, *Z* = 3.032, *p* = 0.002 (2-tailed), *r* = 0.357). Regarding the obtained results, it can be stated that the average fixation number was significantly less with a medium effect in the case of the XAML: *Mdn* = 482 than in the case of the C#: *Mdn* = 661.

The distribution of the average fixation number based on XAML, and classic C# is shown in Figure 10.

In response to the RQ2 question, a significant difference was found in the average fixation number (*p* = 0.002) when the test subjects read and understood the source codes based on XAML or classic C#. The results show that less fixation number, i.e., a focus point number or recording of information, was necessary in the case of the XAML-based UI description.

#### 4.3.3. Average of Pupil Diameter Based on AOIs

The descriptive statistical table of the recorded average pupil diameter results can be seen in Table 4 for XAML and classic C#.

Figure 11 shows the relationship between the average pupil diameter eye movement parameter recorded while reading and understanding the XAML and classic C# source codes.

The normality results of the average pupil diameter measured in the XAML code are significant (XAML: *W*(36) = 0.918, *p* = 0.011, C#: *W*(36) = 0.963, *p* = 0.269). Due to the results of the normality test, the Wilcoxon signed-rank test was used: (*T* = 377.5, *Z* = 1.025, *p* = 0.305 (2-tailed), *r* = −0.121). Regarding the obtained results, it can be stated that the average pupil diameter was not significantly smaller in the case of the XAML: *Mdn* = 46.5 than in the case of the C#: *Mdn* = 48.5.

The distribution of the average pupil diameter based on XAML, and classic C# is shown in Figure 12.

In response to the RQ2 question, a significant difference was not found in the average pupil diameter (*p* = 0.305) when the test subjects read and understood the source codes based on XAML or classic C#.

The more modern XAML technology was basically created due to the development of the GUI of the more efficient and lighter desktop application, which is also confirmed by the results of the knowledge level assessment used in the research. Furthermore, based on the results obtained, it can be stated that a correlation can be demonstrated in the results of the eye movement parameters and the knowledge level assessment tests. This means that eye-tracking analysis can also be used as an alternative knowledge-level assessment. With this objective measurement method, the teacher can make sure that a student’s knowledge is lacking or that it exists.

## 5. Discussion

Based on the recorded and complete knowledge level assessment results, the subjective opinions given on the questionnaires, and the eye movement parameters, it can be stated that XAML-based UI descriptions give better test results and can be read and understood with a lower cognitive and mental load than those based on classic C#. The test subjects’ average information recording, i.e., fixation duration, was significantly longer, and the number of focus points, i.e., information recording and fixation points, was significantly higher when understanding classic C#-based UI descriptions. The recording and processing of significantly less and shorter information showed a more efficient processing of the information located at the given position in XAML-based UI descriptions. In addition to examining fixations, the examination of changes in pupil diameter is widely used to analyze cognitive load; however, in the current research, no significant difference between the two methods, and only a small difference was detected.

A relationship can be shown between the knowledge level assessment test results, subjective opinions, and eye movement parameters because, in the research, a better test result meant a lower fixation duration and fewer fixation points. The opinions of the test subjects also clearly show that XAML-based source codes are easier to understand, and there is no real need for comments to support understanding.

Based on these results, it can be stated that in the field of education, the use of such eye movement tracking devices can be used as an additional measurement method in addition to traditional test and questionnaire results [36,37]. In addition, in the event that the eye movement parameters show greater uncertainty for the students, the teacher can advise repeating the course material and re-measure their understanding or possibly using other measurement methods [38,39]. In addition, the development and continuous renewal of newer methods, self-learning [40] for measuring cognitive load, is becoming more and more necessary nowadays [41,42].

The current research results show that it is advisable to teach .NET-based WPF UI development in XAML because it can be conducted with greater efficiency, thus increasing the efficiency of learning. In addition, it can be used to effectively separate the UI and the business logic, which can result in more efficient team-based development.

In addition to all this, it may be worthwhile to compare the individual technologies in the field of programming education and examine them with eye movement tracking devices, as the revealed results may also affect the organization of education. Introducing easier-to-learn tools, techniques, and methodologies at the beginning of the training can ultimately result in more effective knowledge transfer later on when the students have to learn the more complicated part of the curriculum.

In addition to education, the revealed results can also affect UI software development methods used in the industry, as they can indicate the importance of choosing appropriate, personalized, or team-specific development tools, techniques, and methods.

With the development of devices that track eye movement and the refinement of measurement methods, the use of these devices, in addition to education and programming, may appear in more and more fields in which they are not used at all or less so today.

## 6. Limitations and Future Directions

The results of the current research can be considered with their limitations. Limitations of the research can be considered the low number of questions examining subjective opinions, as well as the similarity and low standard deviation of the age of the selected test subjects. In the future, it would be advisable to extend the study to other universities and test subjects with different experiences and ages. In such tests, not only two software tools could be analyzed, but also several options supporting the creation of a modern GUI. In addition to all this, the use of additional HCI tools, e.g., BCI (Brain Computer Interface), would mean additional research potential, and in addition, an even more precise objective knowledge level assessment could be possible. After solving the ethical problems, the analysis of the patterns of eye movement parameters of the test subjects suffering from cognitive problems while reading and understanding the source codes may be an additional possibility.

Focusing on the field of programming, the analysis of the eye movement parameters of Front-End, Back-End, and Full-Stack developers can contain excellent research opportunities and results because, based on these parameters, different visual patterns could be observed and explored, and further cognitive processes could be analyzed. In addition to all this, on the Front-End side, by examining the eye movement parameters of the different user interfaces, it is possible to analyze their efficiency and manageability, and they can also point out the specific problems of the interface, thus helping the developers to create a user-friendly interface.

## 7. Conclusions

Technologies that help human-computer interaction and communication, such as eye movement tracking, will be increasingly used with greater efficiency in the future to examine and analyze complex cognitive processes. The current research examined two UI descriptive languages (XAML and classic C#) with the participation of test subjects and analyzed their readability and interpretability using a traditional knowledge level assessment test, a questionnaire, and the recording and analysis of eye movement parameters. In order to carry out the test, a complex IT test system was created, with the help of which the necessary parameters could be recorded, and the results were persistently stored for further evaluation. The article emphasizes the importance of selecting programming tools, techniques, technologies, and methods personalized to the person or development team based on the results determined. With the right toolkit, developers can create applications that are more efficient, easier to maintain, and more developable.

## Figures and Tables

**Figure 1 sensors-23-03823-f001:**
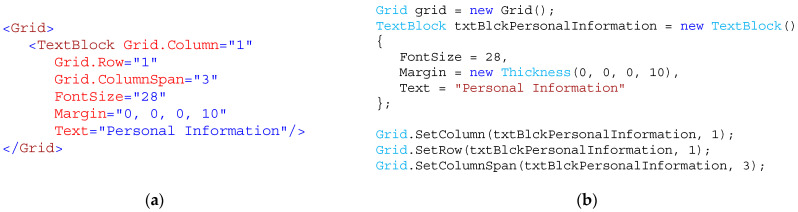
XAML; (**a**) and classic C# (**b**) source code is semantically identical.

**Figure 2 sensors-23-03823-f002:**
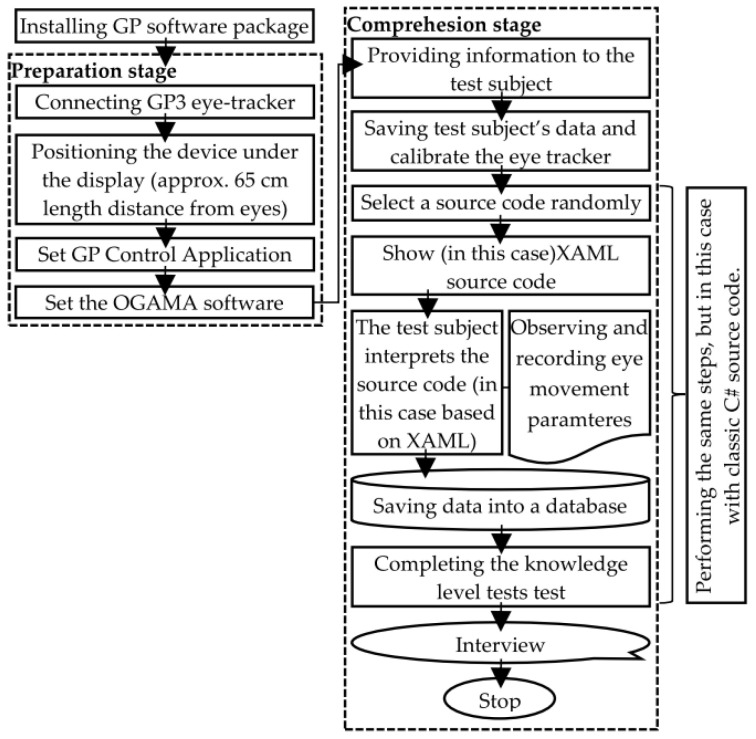
A schematic diagram of the equipment setup.

**Figure 3 sensors-23-03823-f003:**
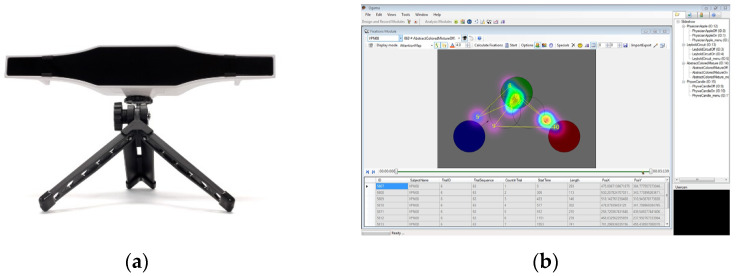
GP3 eye-tracker device; (**a**) and the recording module of OGAMA software package (**b**).

**Figure 4 sensors-23-03823-f004:**
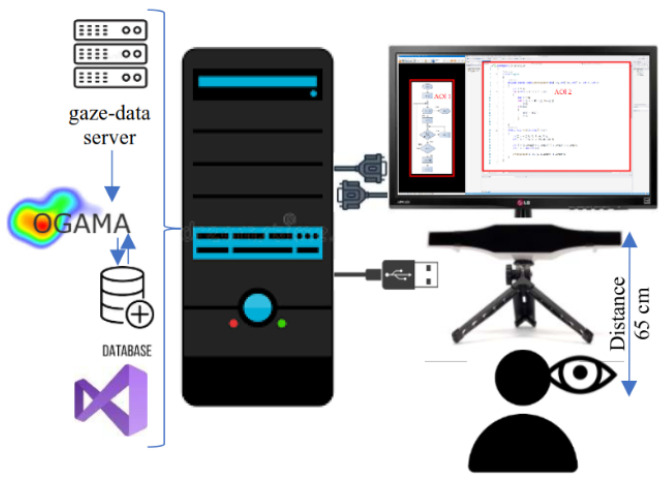
A schematic diagram of equipment setup (Reprinted from ref. [35]).

**Figure 5 sensors-23-03823-f005:**
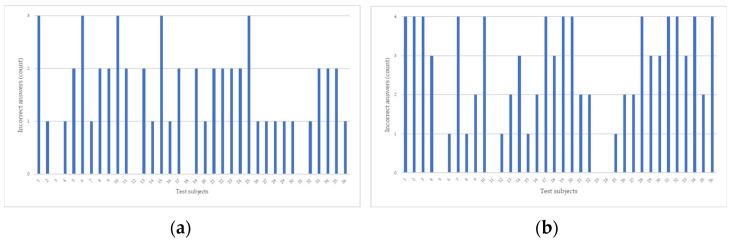
Distribution of incorrect answer (count) based on XAML (**a**) and classic C# (**b**).

**Figure 6 sensors-23-03823-f006:**
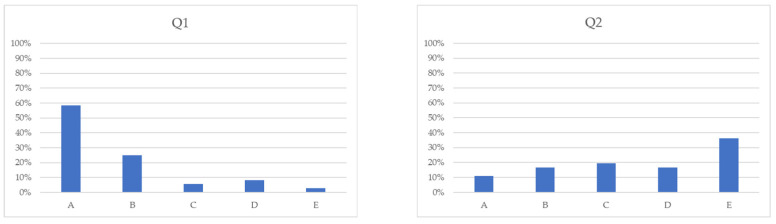
Distribution of responses to the questionnaire.

**Figure 7 sensors-23-03823-f007:**
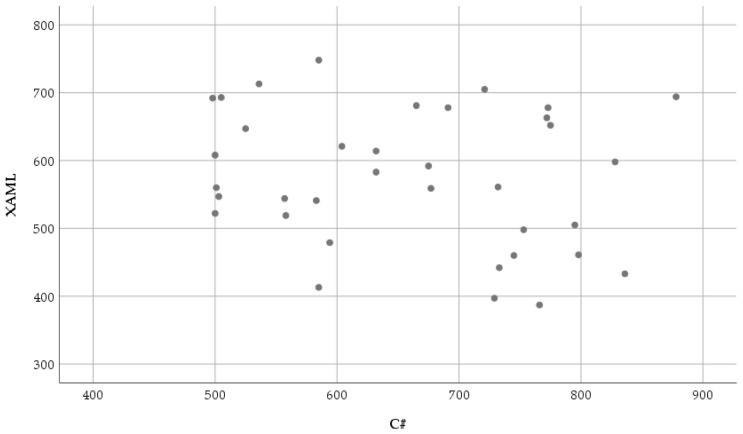
Scatter plot of the fixation duration mean (milliseconds) based on XAML and classic C#.

**Figure 8 sensors-23-03823-f008:**
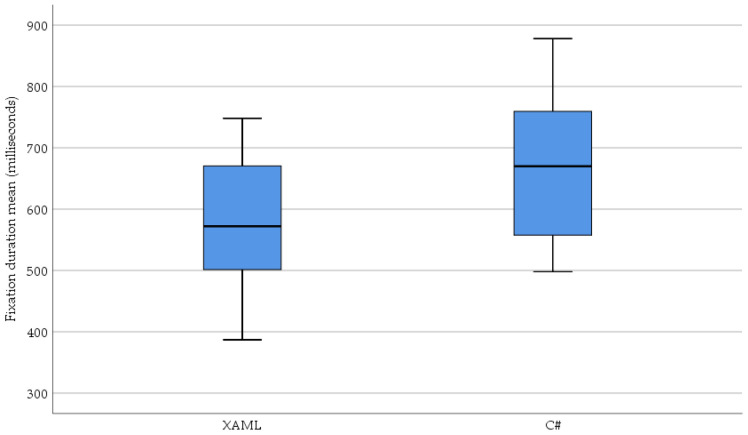
The distribution of the fixation duration mean (milliseconds) is based on XAML and classic C#.

**Figure 9 sensors-23-03823-f009:**
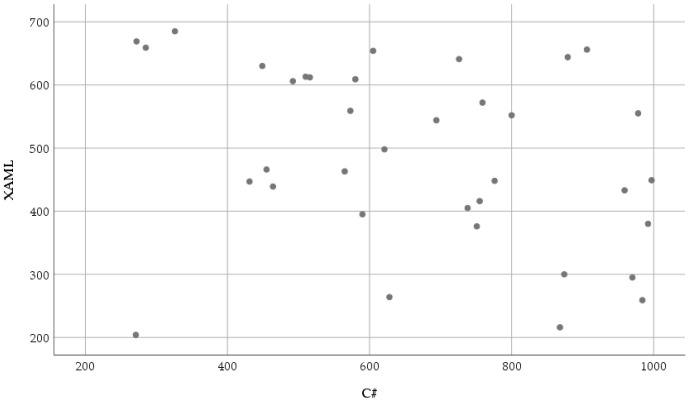
Scatter plot of the number of fixations (count) based on XAML and classic C#.

**Figure 10 sensors-23-03823-f010:**
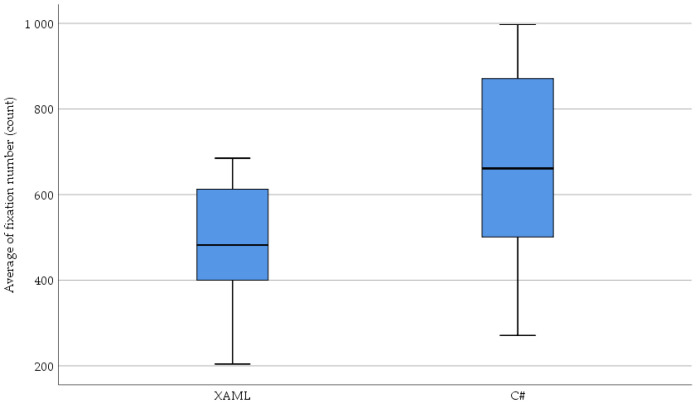
The distribution of the number of fixations (count) based on XAML and classic C#.

**Figure 11 sensors-23-03823-f011:**
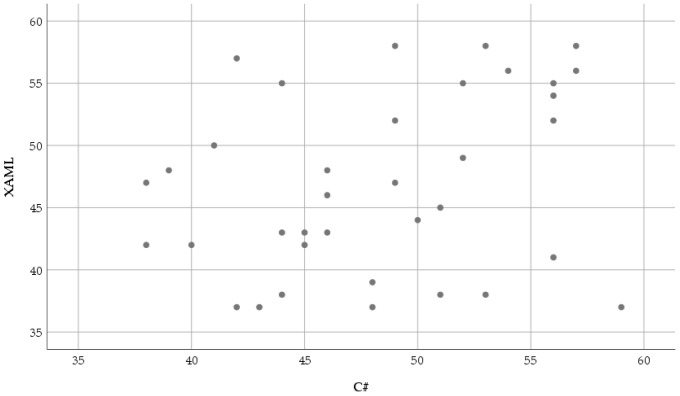
Scatter plot of the average pupil diameter (pixels) based on XAML and classic C#.

**Figure 12 sensors-23-03823-f012:**
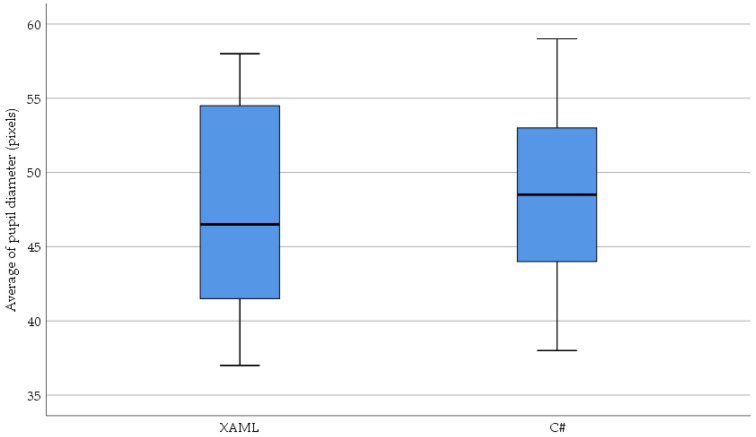
The distribution of the average pupil diameter (pixels) based on XAML and classic C#.

**Table 1 sensors-23-03823-t001:** A summary table on the number of incorrect answers (possible max: 10, possible min: 0) regarding the XAML and classic C# code (*n* = 36).

XAML (Count)				Classic C# (Count)			
Min	Max	Mean	SD	Min	Max	Mean	SD
0	3	1.56	0.88	0	4	2.53	1.40

**Table 2 sensors-23-03823-t002:** The descriptive statistic of the fixation duration mean is based on XAML and classic C# (*n* = 36).

XAML (Milliseconds)				Classic C# (Milliseconds)			
Min	Max	Mean	SD	Min	Max	Mean	SD
387	748	574.67	100.21	498	878	659.45	114.94

**Table 3 sensors-23-03823-t003:** The descriptive statistic of the number of fixations based on XAML and classic C# (*n* = 36).

XAML (Count)				Classic C# (Count)			
Min	Max	Mean	SD	Min	Max	Mean	SD
204	685	489.25	140.82	271	997	667.75	221.61

**Table 4 sensors-23-03823-t004:** The descriptive statistic of the average pupil diameter is based on XAML and classic C# (*n* = 36).

XAML (Pixels)				Classic C# (Pixels)			
Min	Max	Mean	SD	Min	Max	Mean	SD
37	58	46.86	7.19	38	59	48.30	5.97

## Data Availability

Not applicable.

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
