# Peer review of "An Eye Movement Study in Unconventional Usage of Different Software Tools"

_sensors, 2023, doi:10.3390/s23083823_

Round 1

Reviewer 1 Report

The research is well organized, conceptualization is adequate and related content is designed ingeniosuly. As recommendations to stress the quality of the paper and polish the results the suggestions to be considered might be found hereinafter,

1.On what basis the width of respondent audience was determined and the expertise levels, relatedness and demographics of respondents should be clarified and presented in details to the readers. Please perform a literature research on the respondent audience sample size determination techniques, and also, verify the suitability of the sample size determined and used for the study by basing it on reliable studies in the literature.

2.What is the motivation of the proposed work? XAML technology developed by Microsoft is available from Microsoft's web page (https://learn.microsoft.com/en-us/dotnet/desktop/wpf/xaml/?view=netdesktop-7.0) and a very popular web page such as tutorials point looking at (https://www.tutorialspoint.com/xaml/xaml_quick_guide.htm#:~:text=XAML%20stands%20for%20Extensible%20Application,mainly%20used%20for%20designing%20GUIs.) visual objects and references of the visual interface It is seen that it has been developed with the aim of designing it in a simple way. Although some software developers, who are just used to writing code, tend to work on code, the popularity of concepts such as micro service and micro frontend shows that the expertise has changed. For this reason, XAML-style (html-like) methods developed for people who specialize in concepts such as UX visually rather than code are already accepted by experts as providing general convenience in GUI development. The study is useful in terms of the technology and method used, but it is useless in terms of the scenario used. The study can also be positive in that it shows that it is the right decision to separate the development and design processes.

3.Overall, the basic background is not introduced well, where the choice of the proposed approach is not illustrated much clear. Authors are suggested to explain why the selected eye tracking tools were selected, and why the questionnaire was developed on only six questions, as well as, is the amount and content of the questions are adequate to assess the validity of the outcomes.

4. It is recommended that the authors present a detailed comparison of this study with other existing studies developed with similar handling approaches on similar topics i.e. analysis on the usability and performance benchmarking of varying coding languages. What is the difference between this study and existing other studies that apply similar methods to the similar subjects, what contributions this study makes to the decision science considering the other studies. Justify it and give appropriate explanation to do so in the paper.

5. As a recommendation for the authors; In terms of ease of use on users of similar systems with a uniform full stack Dev and separate design development processes (backend/frontend dev), a comparison study using similar methods would be more beneficial.

Author Response

Sensors Paper Review Form

Title of the paper: A comparison between classic C# and XAML codes based on eye-tracking parameters

Manuscript ID: sensors-2238358

I would like to thank the reviewers for their thoughtful comments and efforts towards improving my manuscript. In the following paragraphs, I highlight some general points of the review form, and then I address comments specifically made by the reviewer.

General Reviewer’s Points

General reviewer’s comment

The research is well organized, conceptualization is adequate and related content is designed ingeniosuly.

Author’s answer/comment/response:

I would like to thank you for your positive feedback, which has contributed to improving the quality and overall clarity of my paper.

Response to Reviewer’s Comments

Reviewer Comment 1:
On what basis the width of respondent audience was determined and the expertise levels, relatedness and demographics of respondents should be clarified and presented in details to the readers. Please perform a literature research on the respondent audience sample size determination techniques, and also, verify the suitability of the sample size determined and used for the study by basing it on reliable studies in the literature.

Author’s answer/comment/response:

Thank you very much for your instructive feedback. Using sample size calculation, we can decide what is the minimum number of participants necessary to achieve a given power.

Applied data:

Test family: Paired simple t-test or Wilcoxon signed-rank test

Sample groups: Same subjects

Number of tails: Two

Significance level (α): 0.05

Power: 0.8 (Statistical power is the ability of study to detect a result that is exists in nature. Generally, we want power to be as high as possible. A value of 0.8 is often used in practice.)

The necessary number of participants to achieve this power is: 34.

However, in the future I will examine these research questions with more test subjects in the future.

Reviewer Comment 2:
What is the motivation of the proposed work? XAML technology developed by Microsoft is available from Microsoft's web page (https://learn.microsoft.com/en-us/dotnet/desktop/wpf/xaml/?view=netdesktop-7.0) and a very popular web page such as tutorials point looking at (https://www.tutorialspoint.com/xaml/xaml_quick_guide.htm#:~:text=XAML%20stands%20for%20Extensible%20Application,mainly%20used%20for%20designing%20GUIs.) visual objects and references of the visual interface It is seen that it has been developed with the aim of designing it in a simple way. Although some software developers, who are just used to writing code, tend to work on code, the popularity of concepts such as micro service and micro frontend shows that the expertise has changed. For this reason, XAML-style (html-like) methods developed for people who specialize in concepts such as UX visually rather than code are already accepted by experts as providing general convenience in GUI development. The study is useful in terms of the technology and method used, but it is useless in terms of the scenario used. The study can also be positive in that it shows that it is the right decision to separate the development and design processes.

Author’s answer/comment/response:

Thank you very much for your instructive feedback. My motivation can be traced back to two things. The first is that those students who only put together the graphic interfaces using the drag&drop method often do not understand what code is generated in the background, which means that if something needs to be improved at the source code level, they get stuck and do not understand what is written there. The second is that in most cases these methods pollute the source code and generate more code than is absolutely necessary.

Reviewer Comment 3:
Overall, the basic background is not introduced well, where the choice of the proposed approach is not illustrated much clear. Authors are suggested to explain why the selected eye tracking tools were selected, and why the questionnaire was developed on only six questions, as well as, is the amount and content of the questions are adequate to assess the validity of the outcomes.

Author’s answer/comment/response:

The device and the software package used are cost-effective and several researcher have already used them successfully for previous research [32-34]. Also, I created a figure (figure 2) to represent the testing process. The number of questions is sufficient to discover a kind of similarity between the knowledge level assessment results, eye movement parameters, and subjective opinions, however, in terms of validation, it may be more appropriate to rely on objective data. I asked the test subjects the questions that might be most relevant to the research. The primary goal of the research was to show that eye movement tracking systems can be used to assess the understanding of source code. Of course, the number of questions could be increased, with which further connections could be explored. I will certainly apply the suggestion during my next research.

Reviewer Comment 4:
It is recommended that the authors present a detailed comparison of this study with other existing studies developed with similar handling approaches on similar topics i.e. analysis on the usability and performance benchmarking of varying coding languages. What is the difference between this study and existing other studies that apply similar methods to the similar subjects, what contributions this study makes to the decision science considering the other studies. Justify it and give appropriate explanation to do so in the paper.

Author’s answer/comment/response:

I couldn't find any research that would have previously analyzed the source codes (XAML and classic C#) in the larger databases. The novelty of the current IT-based eye movement tracking system compared to similar systems is that the eye movement parameters were compared with traditional cognitive level assessment tests and were not only evaluated by themselves.

Reviewer Comment 5:
As a recommendation for the authors; In terms of ease of use on users of similar systems with a uniform full stack Dev and separate design development processes (backend/frontend dev), a comparison study using similar methods would be more beneficial.

Author’s answer/comment/response:

Sincerely, thank you very much for the recommendation. I found it very interesting and I intend to deal with it in every way in the future.

Reviewer 2 Report

The authors compared two languages supporting UI implementation related to .NET technology; XAML and classic C# in terms of the cognitive load of test subjects. The ideas in the paper are very interesting and the paper is well writen, with some minor errors listed as follows. The next comments intend to help improving the paper quality and readers' understanding of the paper.

I believe my main concern about the paper regards its "obvious" results obtained from the experiments performed. The authors find that XAML is easier to read and understand then the same description in classic C#. What happens is that .net framework dates from 2002, while XAML is about 8 years yonger (it dates from 2010). It is easy to assume that XAML is better than classic C#, otherwise what would justify it being used by developers?

It would be nice to compare classic C# and XAML also with the visual development of the UI. Nowadays, the developer may use the WYSIWYG approach (what you see is what you get) and developt a UI by completely using visual tools. It is hard to find a developer that creates a UI interface from stracth without using the visual tools to implement a UI. Even if the visual tools are used just as a start on the development of the UI, they usually make the process of UI creation faster. I believe the use of visual tools should be added to the comparison as a reference time implementation, as I believe it is faster to create/modify the UI by clicking and dragging components instead of writing code (XAML or classic C#).

"The evaluation of the results began with the evaluation of traditional knowledge level assessment tests, which examined the comprehensibility and readability of the studied source code. Each test subject received the same 10 questions, so a total of 360 answers were evaluated. " -> it would be nice to have more detail regarding the 10 questions used in this case

It would be interesting to also add to the comparison C# Markup (https://www.linkedin.com/pulse/comparison-between-classic-c-markup-xaml-creating-ui-xamarin-matos/), as it seems to be in the middle between classic C# and XAML (it has the advantages of both worlds).

Please provide more details regarding the experiments performed. Did the 36 students that participated on the tests had to use the 4 UI descriptions? Did you manage to randomly change the order on what UI descriptions users had access first? These factor may influence the result of the experiments. Do the testers filled the questionnaires right after testing a specific configuration of in the end, after testing all possible configurations?

What about the time for the tests? Was the time to perform the test also captured? Please provide more information regarding time since it could be another important parameter used to compare both approaches.

Why is it important that the analysis of the eye done based on the eye tracker is the same or is related to the analysis from the 10 questions?

Please highlight the contributions of the paper. Please be specific and if possible, compare the findings of the work with the state of the art.

It seems that Figure 10 and Figure 11 show the same information using different visualization approaches. Are both figures really necessary?

Table 4 should be equivalent to Figure 11, but they are not. Did I miss anything?

More general comments and minor errors are listed as follows.

"The language plays a particularly important role in the .NET framework, and it can be especially important in the development of WPF (Windows Presentation Foundation) or Xamarin applications, where the UI data elements, data bindings and other services, etc. defines." -> ?

"code. [1] " -> "code [1]. "

"Chapter" -> "Section" (please change this for all instances in the paper)

"last chapter." -> "last section."

"Lew et al. [19] operators" -> "Lew et al. [19] showed that operators"

"As the goal of the current research already stated above, he wants to investigate which language the use of means a lower cognitive load. " -> please rewrite

"during his examination. " -> "during this examination. "

"Table I. summary table " -> "Table 1"

"In addition to all this, it can also be seen based on the subjective opinions that the test subjects preferred and easier to interpret the XAML-based codes, on the contrary, with the classic C# code." -> please rewrite

"shown in Figure 9." -> "is shown in Figure 9."

"shown in Figure 11." -> "is shown in Figure 11."

"was significantly more" -> "was significantly higher"

"Because when using tools that better match the competence of the development team, you can create more readable and easier-to-interpret source code, which can result in a more efficiently maintainable application. " -> please rewrite

Author Response

Sensors Paper Review Form

Title of the paper: A comparison between classic C# and XAML codes based on eye-tracking parameters

Manuscript ID: sensors-2238358

I would like to thank the reviewers for their thoughtful comments and efforts towards improving my manuscript. In the following paragraphs, I highlight some general points of the review form, and then I address comments specifically made by the reviewer.

General Reviewer’s Points

General reviewer’s comment

The authors compared two languages ​​supporting UI implementation related to .NET technology; XAML and classic C# in terms of the cognitive load of test subjects. The ideas in the paper are very interesting and the paper is well written, with some minor errors listed as follows. The following comments intend to help improve the paper quality and readers' understanding of the paper.

Author’s answer/comment/response:

I would like to thank you for your positive feedback, which has contributed to improving the quality and overall clarity of my paper.

Response to Reviewer’s Comments

Reviewer Comment 1:
I believe my main concern about the paper regards its "obvious" results obtained from the experiments performed. The authors find that XAML is easier to read and understand then the same description in classic C#. What happens is that .net framework dates from 2002, while XAML is about 8 years yonger (it dates from 2010). It is easy to assume that XAML is better than classic C#, otherwise what would justify it being used by developers?

Author’s answer/comment/response:

Thank you very much for your instructive feedback. The outcome of the results could be assumed, however, the primary goal of the article was to examine the evolution of the eye movement parameters also with regard to the traditional knowledge level assessment results. Indeed, XAML proved to be more effective, which was also supported by the eye movement parameters. With regard to the obtained result, it can be stated that eye movement tracking can be used as an objective examination procedure and can also appear as a kind of learning support system.

Reviewer Comment 2:
It would be nice to compare classic C# and XAML also with the visual development of the UI. Nowadays, the developer may use the WYSIWYG approach (what you see is what you get) and developt a UI by completely using visual tools. It is hard to find a developer that creates a UI interface from stracth without using the visual tools to implement a UI. Even if the visual tools are used just as a start on the development of the UI, they usually make the process of UI creation faster. I believe the use of visual tools should be added to the comparison as a reference time implementation, as I believe it is faster to create/modify the UI by clicking and dragging components instead of writing code (XAML or classic C#).

Author’s answer/comment/response:

Thank you very much for your instructive feedback. My motivation can be traced back to two things. The first is that those students who only put together the graphic interfaces using the drag&drop method often do not understand what code is generated in the background, which means that if something needs to be improved at the source code level, they get stuck and do not understand what is written there. The second is that in most cases these methods pollute the source code and generate more code than is absolutely necessary. However, thank you very much for the suggestion. In the future, I definitely plan to do such a comparison.

Reviewer Comment 3:
It would be interesting to also add to the comparison C# Markup (https://www.linkedin.com/pulse/comparison-between-classic-c-markup-xaml-creating-ui-xamarin-matos/), as it seems to be in the middle between classic C# and XAML (it has the advantages of both worlds).

Author’s answer/comment/response:

Thank you very much for the suggestion. I would definitely like to deal with it in a future research.

Reviewer Comment 4:
Please provide more details regarding the experiments performed. Did the 36 students that participated on the tests had to use the 4 UI descriptions? Did you manage to randomly change the order on what UI descriptions users had access first? These factor may influence the result of the experiments. Do the testers filled the questionnaires right after testing a specific configuration of in the end, after testing all possible configurations?

Author’s answer/comment/response:

Thank you very much for your instructive feedback. The users were randomly assigned one XAML and one classic C# source code, thus avoiding that the prior knowledge of the test subjects would influence the result. Each knowledge level assessment test had to be completed after studying each type of source code. I added a diagram (figure 2) to the article, which can help you better understand the process.

Reviewer Comment 5:
What about the time for the tests? Was the time to perform the test also captured? Please provide more information regarding time since it could be another important parameter used to compare both approaches.

Author’s answer/comment/response:

Thank you very much for your instructive feedback. Time didn't matter in this test environment, as I didn't want it to affect knowledge. The time factor can be taken into account during a subsequent investigation, thank you for your suggestion.

Reviewer Comment 6:
Why is it important that the analysis of the eye done based on the eye tracker is the same or is related to the analysis from the 10 questions?

Author’s answer/comment/response:

Thank you very much for your instructive feedback. The primary goal of the article was to examine the evolution of the eye movement parameters also with regard to the traditional knowledge level assessment results.

Reviewer Comment 7:
Please highlight the contributions of the paper. Please be specific and if possible, compare the findings of the work with the state of the art.

Author’s answer/comment/response:

Thank you very much for your instructive feedback. I couldn't find any research that would have previously analyzed the source codes (XAML and classic C#) in the larger databases. The novelty of the current IT-based eye movement tracking system compared to similar systems is that the eye movement parameters were compared with traditional cognitive level assessment tests and were not only evaluated by themselves.

Reviewer Comment 8:
It seems that Figure 10 and Figure 11 show the same information using different visualization approaches. Are both figures really necessary?

Author’s answer/comment/response:

Thank you very much for your instructive feedback. The two chart types use different approaches to represent data. Of course, this is annoying, it can be deleted.

Reviewer Comment 9:
Table 4 should be equivalent to Figure 11, but they are not. Did I miss anything?

Author’s answer/comment/response:

I checked the table and Figure 11, but found them to be the same. Please describe exactly where you see a discrepancy.

Reviewer Comment 10:
"The language plays a particularly important role in the .NET framework, and it can be especially important in the development of WPF (Windows Presentation Foundation) or Xamarin applications, where the UI data elements, data bindings and other services, etc. defines." -> ?

"code. [1] " -> "code [1]. "

"Chapter" -> "Section" (please change this for all instances in the paper)

"last chapter." -> "last section."

"Lew et al. [19] operators" -> "Lew et al. [19] showed that operators"

"As the goal of the current research already stated above, he wants to investigate which language the use of means a lower cognitive load. " -> please rewrite

"during his examination. " -> "during this examination. "

"Table I. summary table " -> "Table 1"

"In addition to all this, it can also be seen based on the subjective opinions that the test subjects preferred and easier to interpret the XAML-based codes, on the contrary, with the classic C# code." -> please rewrite

"shown in Figure 9." -> "is shown in Figure 9."

"shown in Figure 11." -> "is shown in Figure 11."

"was significantly more" -> "was significantly higher"

"Because when using tools that better match the competence of the development team, you can create more readable and easier-to-interpret source code, which can result in a more efficiently maintainable application. " -> please rewrite

Author’s answer/comment/response:

Thank you very much for your instructive feedback. I have changed them.

Round 2

Reviewer 1 Report

The research is not developed enough according to the former suggestions, as reviewer recommendation it should be re-considered and re-structured according to the points found hereinafter,

1.The authors suggest that the study was developed to analyze the student audience who uses the investigated software tools different from the basis they put forward in the earlier version. They appended that information as a answer to reviewer suggestions, and, it is well appreciated. However, the title of the article and abstract parts have also to be re-organized and re-written to address this information and content of the basis of this research.

2. However it is suggetsed that the focus auidinece is student group, the respondent audience was not determined as students. The respondent group ans hence all of the analyze data and results have to be re-considered. The grades, years of expertise, departments, relatedness and demographics of respondents should be clarified and presented in details to the readers.

3. It is appreciated that the authors performed a literature research on the respondent audience sample size determination techniques, and also, verified the suitability of the sample size determined and used for the study by basing it on reliable studies in the literature.

4. XAML technology was already developed to ease the code writing, hence, the results which emphasise on the user-friendly structure and the convenience presented to the users do not address new findings or present new seminal information to the reader, besides, make the announcement of the known facts. Hence result section has to be developed and the expression of contribution to the literature has to be improved.

5. The issues related to the amount of the questions, the questionnaire and validity tests of the answers, the respondent audience selection, overall basic background of the study motives has to be appended to the manuscript as the limitations of the research and application section.

6. The recommendations to be considered was suggested to the authors to be analyzed and appended particular to this study of them, hence, if they can it will be beneficial for them to perform a comparison study using similar methods in terms of ease of use on users of similar systems with a uniform full stack Dev and separate design development processes (backend/frontend dev), and if they decide not to despite the value it could add to their study, they are advised to address and explain the mentioned research as future works related to the study.

Author Response

Sensors Paper Review Form

Title of the paper: A comparison between classic C# and XAML codes based on eye-tracking parameters

Manuscript ID: sensors-2238358

I would like to thank the reviewers for their thoughtful comments and efforts towards improving my manuscript. In the following paragraphs, I highlight some general points of the review form, and then I address comments specifically made by the reviewer.

Response to Reviewer’s Comments

Reviewer Comment 1:
The authors suggest that the study was developed to analyze the student audience who uses the investigated software tools different from the basis they put forward in the earlier version. They appended that information as a answer to reviewer suggestions, and, it is well appreciated. However, the title of the article and abstract parts have also to be re-organized and re-written to address this information and content of the basis of this research.

Author’s answer/comment/response:

Thank you very much for your instructive feedback. I have changed the title of the manuscript and re-organized and added the information to the abstract.

Reviewer Comment 2:
However it is suggetsed that the focus auidinece is student group, the respondent audience was not determined as students. The respondent group ans hence all of the analyze data and results have to be re-considered. The grades, years of expertise, departments, relatedness and demographics of respondents should be clarified and presented in details to the readers.

Author’s answer/comment/response:

In the article the respondent audience was determined as student. The test was attended by 36 university students (10 woman and 26 man, age: M=20.25 SD=1.05), who successfully completed the subjects containing the study materials, the knowledge of which is essential for solving the tests. The test subjects volunteered for the test and declared themselves to be completely healthy. They were not under the influence of any medication and had no difficulties in reading or learning in the past and during this examination. Based on the results of the programming subject, test subjects with better than average programming skills and a similar level of knowledge were selected.

Reviewer Comment 3:
It is appreciated that the authors performed a literature research on the respondent audience sample size determination techniques, and also, verified the suitability of the sample size determined and used for the study by basing it on reliable studies in the literature.

Author’s answer/comment/response:

Thank you very much.

Reviewer Comment 4:
XAML technology was already developed to ease the code writing, hence, the results which emphasise on the user-friendly structure and the convenience presented to the users do not address new findings or present new seminal information to the reader, besides, make the announcement of the known facts. Hence result section has to be developed and the expression of contribution to the literature has to be improved.

Author’s answer/comment/response:

Thank you for your instructive comment. It was added to the result section:

The more modern XAML technology was basically created due to the development of the GUI of the more efficient and lighter desktop application, which is also confirmed by the results of the knowledge level assessment used in the research. Furthermore, based on the results obtained, it can be stated that a correlation can be demonstrated in the results of the eye movement parameters and the knowledge level assessment tests. This means that eye tracking analysis can also be used as an alternative knowledge level assessment. With this objective measurement method, the teacher can make sure that a student's knowledge is lacking or that it exists.

Reviewer Comment 5:
The issues related to the amount of the questions, the questionnaire and validity tests of the answers, the respondent audience selection, overall basic background of the study motives has to be appended to the manuscript as the limitations of the research and application section.

Author’s answer/comment/response:

Thank you for your instructive feedback. I wrote a limitations and future directions section.

Reviewer Comment 6:
The recommendations to be considered was suggested to the authors to be analyzed and appended particular to this study of them, hence, if they can it will be beneficial for them to perform a comparison study using similar methods in terms of ease of use on users of similar systems with a uniform full stack Dev and separate design development processes (backend/frontend dev), and if they decide not to despite the value it could add to their study, they are advised to address and explain the mentioned research as future works related to the study.

Author’s answer/comment/response:

Thank you for your instructive feedback. I wrote about your suggestion in the limitations and future directions section.

Reviewer 2 Report

Thanks for the revised version of the paper, I believe it can be accepted now.

Sorry for the comparison between table 4 and figure 11, I was mistaken.

Author Response

Sensors Paper Review Form

Title of the paper: A comparison between classic C# and XAML codes based on eye-tracking parameters

Manuscript ID: sensors-2238358

I would like to thank the reviewers for their thoughtful comments and efforts towards improving my manuscript.

General Reviewer’s Points

 General reviewer’s comment

Thanks for the revised version of the paper, I believe it can be accepted now.

Sorry for the comparison between table 4 and figure 11, I was mistaken.

Author’s answer/comment/response:

I would like to thank you for your positive feedback, which has contributed to improving the quality and overall clarity of my paper.
